# LatentAugment: Dynamically Optimized Latent Probabilities of Data Augmentation

## Abstract

Although data augmentation is a powerful technique for improving the performance of image classification tasks, it is difficult to identify the best augmentation policy. The optimal augmentation policy, which is the latent variable, cannot be directly observed. To address this problem, this study proposes *LatentAugment*, which estimates the latent probability of optimal augmentation. The proposed method is appealing in that it can dynamically optimize the augmentation strategies for each input and model parameter in learning iterations. Theoretical analysis shows that LatentAugment is a general model that includes other augmentation methods as special cases, and it is simple and computationally efficient in comparison with existing augmentation methods. Experimental results show that the proposed LatentAugment has higher test accuracy than previous augmentation methods on the CIFAR-10, CIFAR-100, SVHN, and ImageNet datasets.

Data augmentation is a widely used technique for generating additional data to improve the performance of computer vision tasks (Shorten & Khoshgoftaar, 2019). Although data augmentation performs well in experimental studies, designing data augmentations requires human expertise with prior knowledge of the dataset, and it is often difficult to transfer the augmentation strategies across different datasets (Krizhevsky et al., 2012). Recent studies on data augmentation consider an automated design process of searching for augmentation strategies from a dataset. For example, AutoAugment, proposed by Cubuk et al. (2018), uses reinforcement learning to automatically explore data augmentation policies using smaller network models and reduced datasets. Although AutoAugment shows great improvement on image classification tasks of different datasets, it requires thousands of GPU hours to search for augmentation strategies. Furthermore, the data augmentation operations optimized for reduced datasets using smaller network models may not be optimal for full datasets using larger network models.

To address this problem, this study proposes **LatentAugment**, which estimates the latent probability of the optimal augmentation customized to each input image and network model. There is no doubt that an optimal augmentation policy exists for each input image using a specific network model. However, the optimal augmentation policy, which is a latent variable, cannot be directly observed. Although a latent variable itself cannot be observed , we can estimate the probability of the latent variable being the optimal augmentation policy. LatentAugment applies Bayes' rule, to estimate the conditional probability of the augmentation policy, given the input data and network parameters.

Figure 1 shows the concept of the proposed latent augmentation method. Following the Bayesian data augmentation proposed by Tran et al. (2017), LatentAugment uses the expectation-maximization (EM) algorithm to update the model parameters. In the expectation (E)-step, the expectation of the weighted loss function is calculated using the conditional probability of the latent augmentation policies. In the maximization (M)-step, the expected loss function is minimized using the standard stochastic gradient descent. The conditional probabilities of the highest loss function with the augmentation policy were calculated using the loss function with the updated parameters and input data. The unconditional probabilities of the augmentation policies were generated using the moving average of the conditional probabilities. Note that the conditional probabilities of the latent augmentation policies are dynamically optimized for the input and updated model parameters in the iterations of the EM algorithm.

The contribution of this study can be summarized as follows:

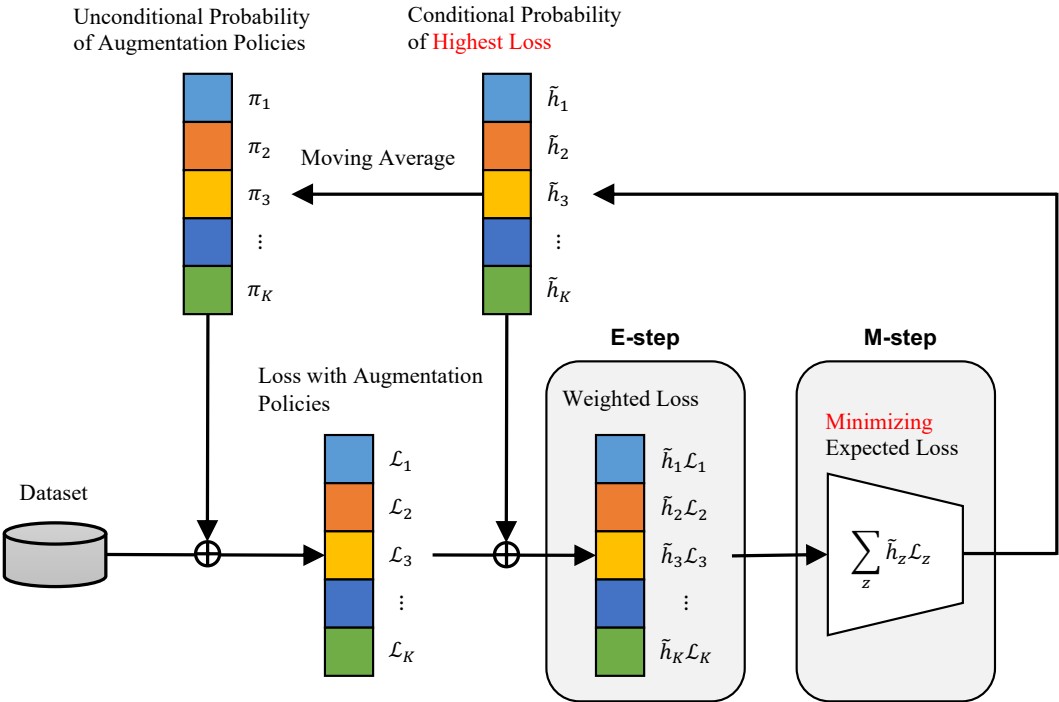

Figure 1: An overview of the proposed LatentAugment. The loss functions with augmentation policies are calculated using the input data and the unconditional probability of augmentation policies. The model parameters are updated by the EM algorithm. In E-step, the expectation of the weighted loss function is calculated using the conditional probability of the highest loss. In M-step, the expected loss function is minimized using the standard stochastic gradient descent. The conditional probabilities of the highest loss are calculated using the loss function with the updated parameters and input data. The unconditional probabilities of the augmentation policies are generated by the moving average of the conditional probability.

- It provides a theoretical model for LatentAugment. This study shows that LatentAugment can dynamically optimize the augmentation methods for each input and model parameter in the learning iterations by calculating the conditional probabilities of the latent augmentation policies. Furthermore, it shows that LatentAugment is a general augmentation model that includes other augmentation methods, such as Adversarial AutoAugment (Zhang et al., 2019) and uncertainty-based sampling (Wu et al., 2020), as special cases.

- LatentAugment is simple and computationally efficient. It does not require the augmentation policies to be searched before training. Adversarial AutoAugment proposes the application of a generative adversarial network (GAN) (Goodfellow et al., 2014) to solve the maximization of the minimum loss function, which requires an additional training cost for the adversarial network. In contrast, the proposed LatentAugment can solve this problem using the simple stochastic gradient descent algorithm without an adversarial network.

- Experimental results show that the proposed LatentAugment can improve the test accuracy for the CIFAR-10, CIFAR-100, SVHN, and ImageNet datasets. For example, a test accuracy of 98.72% was achieved with the PyramidNet+ShakeDrop (Han et al., 2017; Yamada et al., 2018) on CIFAR-10, which is a significantly better performance compared to previous augmentation methods.

# 1 RELATED WORKS

Several studies have been conducted on data augmentation methods in the literature on machine learning. Shorten & Khoshgoftaar (2019) provided a comprehensive review of image data augmentation. Recent studies have attempted to automatically identify data augmentation methods. Smart augmentation (Lemley et al., 2017) merges two or more samples from the same class to improve the generalization of a target network. AutoAugment (AA) (Cubuk et al., 2018) applies a recurrent neural network (RNN) as a sample controller to search for the best data augmentation policy using small proxy tasks of randomly drawn images from the training dataset. After identifying the best policy, fixed policies are applied to the training dataset. Population-based augmentation (PBA) (Ho et al., 2019) generates dynamic augmentation policy schedules instead of a fixed augmentation policy. RandAugment (RA) (Cubuk et al., 2019) has a significantly reduced search space and allows training on the target task without a separate proxy task. Fast AutoAugment (Fast AA) (Lim et al., 2019) determines the best augmentation policy using a more efficient search strategy based on density matching. Faster AA (Hataya et al., 2019) uses a differentiable policy search pipeline for data augmentation, which is much faster than previous methods. DADA (Li et al., 2020) also reduces the cost of policy search using a differentiable optimization problem via Gumbel-Softmax, while DeepAA (Zheng et al., 2022) uses a multi-layer data augmentation pipeline.

Adversarial AutoAugment (AdvAA) (Zhang et al., 2019) applies an adversarial network to generate data augmentation. While the training network minimizes the loss, the adversarial network maximizes training loss. Uncertainty-Bases Sampling (UBS) (Wu et al., 2020) generates data augmentation of the highest loss without an adversarial network. As shown in the next section, the loss functions of AdvAA and UBS can be regarded as special cases of LatentAugment proposed in this study. MetaAugment (Zhou et al., 2020) uses an additional augmentation policy network to minimize the weighted losses of augmented training images. DHA (Zhou et al., 2021) uses super and child networks to achieve joint optimization of the data augmentation policy, hyper-parameter, and architecture. In contrast, the proposed LatentAugment does not require any additional network for searching the augmentation policy.

The best augmentation policies are the latent variables that cannot be observed. The expectation-maximization (EM) algorithm was proposed to analyze latent variables (Dempster et al., 1977; McLachlan & Krishnan, 2007; Ng et al., 2012). The EM algorithm estimates parameters using an iterative process of expectation and minimization of the loss function. However, when the dataset is large, it might be difficult to calculate the expectation or minimization of the full dataset. To address the difficulty of working with large datasets, some approaches have been proposed, including the generalized EM algorithm (Dempster et al., 1977), Monte Carlo EM algorithm (Wei & Tanner, 1990; Tanner, 1991), stochastic EM algorithm (Nielsen, 2000), and generalized Monte Carlo EM (Tran et al., 2017). For application to data augmentation, Bayesian data augmentation (Tran et al., 2017) estimates the parameters using the EM algorithm to generate data augmentation using the Bayesian approach. Bayesian data augmentation requires an adversarial network, whereas LatentAugment does not use an adversarial network. Nevertheless, Bayesian data augmentation is mostly related to this study in the application of the EM algorithm to data augmentation.

# 2 METHOD

Consider a classification task with $C$ categories for the $N$ training data points $X = \{x_1, x_2, \cdots, x_N\}$ and labels $Y = \{y_1, y_2, \cdots, y_N\}$. Let $P(y \mid x, \theta)$ denote the predicted probability of the output $y$, given the input $x$ and the parameter $\theta$. Consider that each input is transformed using random data augmentation. Let $\mathbb{S} = \{1, \cdots, S\}$ be the set of augmentation policies, and $z^*(x, \theta)$ be the optimal augmentation policy for the input given the parameter. We cannot directly observe the optimal policy; therefore, $z^*(x, \theta)$ is the latent variable. Let $\pi_z$ be the unconditional probability that the augmentation policy $z$ is applied to the input data. The loss function using the augmented data can be written as $\mathcal{L}(\Theta) = -\mathbb{E}_{(x,y) \sim (X,Y)} \log \left( \sum_{z \in \mathbb{S}} \pi_z P(y \mid o_z(x), \theta) \right)$, where $o_z(x)$ denotes the augmented data using the augmentation policy $z$, $\Theta = \{\theta, \pi\}$ and $\pi = \{\pi_1, \cdots, \pi_S\}$.

## 2.1 Generalized EM Algorithm

The loss function with the latent variable can be minimized using the expectation–maximization (EM) algorithm. The EM algorithm is an iterative procedure used to compute the maximum likelihood estimate in the presence of latent variables (Ng et al., 2012). In the E-step, the expected loss function calculated. In the M-step, the parameter is updated by minimizing the expected loss function. Let $\Theta^{(t)} = \{\theta^{(t)}, \pi^{(t)}\}$ be the parameter and the unconditional probability at iteration $t$ and $h_z^{(t)}(x, y, \Theta^{(t)}, \mathbb{S})$ be the conditional probability of the augmentation probability of policy $z$ for the individual data point $x$ given the label $y$ at iteration $t$. Applying Bayes' rule, we can calculate the conditional probability (McLachlan & Krishnan, 2007):

$$h_z^{(t)}(x, y, \Theta^{(t)}, \mathbb{S}) = \frac{\pi_z^{(t)} P\left(y \,|\, o_z(x), \theta^{(t)}\right)}{\sum\limits_{z \in \mathbb{S}} \pi_z^{(t)} P\left(y \,|\, o_z(x), \theta^{(t)}\right)}.$$

Using $h_z^{(t)}$, as shown in Ng et al. (2012), the expected loss function $\mathcal{E}\left(\Theta | \Theta^{(t)}\right)$ can be written as:

$$\mathcal{E}\left(\Theta | \Theta^{(t)}\right) = -\mathbb{E}_{(x,y) \sim (X,Y)} \left[\sum_{z \in \mathbb{S}} h_z^{(t)} \log(\pi_z) + \sum_{z \in \mathbb{S}} h_z^{(t)} \log\left(P\left(y \,|\, o_z(x), \theta\right)\right)\right]. \quad (1)$$

In the M-step, the parameter $\theta$ and the unconditional probability $\pi_z$ were estimated by minimizing the expected loss function given the conditional probability $h_z^{(t)}$. If solving the minimization problem of $\mathcal{E}\left(\Theta | \Theta^{(t)}\right)$ proves difficult, the generalized EM algorithm proposed by Dempster et al. (1977) can be used to estimate $\Theta^{(t+1)}$, where $\mathcal{E}\left(\Theta^{(t+1)} | \Theta^{(t)}\right) < \mathcal{E}\left(\Theta | \Theta^{(t)}\right)$.

Calculation of $\mathcal{E}\left(\Theta | \Theta^{(t)}\right)$ requires the expectation of possible augmentation policies. When the number of augmentation policies $\mathbb{S}$ is large, the computational burden of $\mathcal{E}\left(\Theta | \Theta^{(t)}\right)$ cannot be neglected. Alternatively, the subset $\mathbb{K}$ which is randomly drawn from the full set $\mathbb{S}$, of the augmentation policies, can be used. Then, the conditional probability $h_z^{(t)}$ can be written as

$$h_z^{(t)}\left(x, y, \Theta^{(t)}, \mathbb{K}\right) = \frac{\pi_z^{(t)} P\left(y \,|\, o_z(x), \theta^{(t)}\right)}{\sum\limits_{z \in \mathbb{K}} \pi_z^{(t)} P\left(y \,|\, o_z(x), \theta^{(t)}\right)}. \quad (2)$$

As shown in the Appendix (A.1), if the subset $\mathbb{K}$ is generated using simple random draws from the full set $\mathbb{S}$, the expected loss function using the subset is equal to that obtained using the full set.

## 2.2 Latent Augmentation Policy

The generalized EM algorithm can estimate the parameter by minimizing the expected loss function using latent variables. However, this may cause an overfitting problem. Following AdvAA, an augmentation policy is applied to maximize the loss function using a harder augmentation policy. Let $\mathcal{L}_z^{(t)} = -\log\left(\pi_z^{(t)} \cdot P\left(y \,|\, o_z(x), \theta^{(t)}\right)\right)$ be the the contribution to the loss function for input $(x, y)$, using the augmentation policy $z$ at iteration $t$. Consider the conditional probability of the latent augmentation policy $\tilde{h}_z^{(t)}$ such that the augmentation policy $z$ has the highest loss in the set of $\mathbb{K}$, using the softmin function:

$$\tilde{h}_z^{(t)} = \Pr[\mathcal{L}_z^{(t)} \geq \mathcal{L}_k^{(t)}, \forall k \in \mathbb{K}] = \Pr[h_z^{(t)} \leq h_k^{(t)}, \forall k \in \mathbb{K}] = \frac{\exp(-h_z^{(t)}/\sigma)}{\sum_{k \in \mathbb{K}} \exp(-h_k^{(t)}/\sigma)}, \quad (3)$$

where $\sigma$ is the inverse scale parameter. Note that, from the definition of $\mathcal{L}_z$, the probability of minimum $h_z$ is equal to the one of maximum $\mathcal{L}_z$. Thus, the softmin function is related to the goal of the LatentAugment, maximization of minimum loss. The proposed LatentAugment can be implemented by the EM algorithm with $\tilde{h}_z^{(t)}$. In the E-step, the LatentAugment calculates the

---

**Algorithm 1** LatentAugment

  **Input:** $(X, Y)$: dataset
  **Require:** $B$: the number of mini-batch, $\mathbb{S}$: the full set of augmentation policies, $S$: the size of augmentation policies, and $\sigma$: the inverse scale.
  **Initialize:** $\pi_z = 1/S$, for $z = \{1, \ldots, S\}$. Initialize the network parameter $\theta^{(0)}$.
  **for** $t = 1, \ldots, B$ **do**
    Randomly draw the subset $\mathbb{K}$ from $\mathbb{S}$.
    Calculate $\tilde{h}_z^{(t)}$ using equation (3).
    E-step: Calculate $\tilde{\mathcal{E}}$ using equation (4)
    M-step: Update the parameter $\theta^{(t)}$ and $\pi^{(t)}$ using equation (5)
  **end for**
  **Return:** $\theta^{(B)}$ and $\pi^{(B)}$

---

expected loss function weighted by the probability of the minimum conditional probability $\tilde{h}_z^{(t)}$, instead of $h_z^{(t)}$:

$$\tilde{\mathcal{E}}\left(\Theta|\Theta^{(t)}\right) = -\mathbb{E}_{(x,y)\sim(X,Y)}\mathbb{E}_{\mathbb{K}\sim\mathbb{S}}\left[\sum_{z\in\mathbb{K}}\tilde{h}_z^{(t)}\log\left(\pi_z\right) + \sum_{z\in\mathbb{K}}\tilde{h}_z^{(t)}\log\left(P\left(y\,|\,o_z\left(x\right),\theta\right)\right)\right]. \quad (4)$$

In the M-step, the parameter is updated by minimizing the expected loss function with fixed $\tilde{h}_z^{(t)}$:

$$\theta^{(t+1)} = \theta^{(t)} - \eta\nabla_\theta\tilde{\mathcal{E}}\left(\Theta|\Theta^{(t)}\right), \pi_z^{(t+1)} = \text{Moving Average of } \frac{\mathbb{E}_{(x,y)\sim(X_B,Y_B)}\tilde{h}_z^{(t)}}{\mathbb{E}_{(x,y)\sim(X_B,Y_B)}\sum_{z\in\mathbb{K}}\tilde{h}_z^{(t)}}, \quad (5)$$

where $(X_B, Y_B)$ is the mini-batch of the input data. This process is iterated until convergence is achieved. The estimation procedure with LatentAugment is summarized as Algorithm 1.

## 2.3 Advantages of the LatentAugment

The proposed LatentAugment has following advantages over the existing augmentation methods:

1. **The weighted augmentation policies are optimized for the individual input.** Most recent studies, such as AA, use randomly drawn policies; however, they do not apply policies that are appropriate for each input data. On the other hand, LatentAugment utilizes randomly drawn policies customized for each input by calculating conditional probabilities for the given input.

2. **It provides a closed-form solution for the probability of optimal augment polices.** LatentAugment can estimate the unconditional probability ($\pi_z$) of optimal augment policies using a closed-form solution (5) of the loss minimization. Thus, LatentAugment does not involve the additional cost of searching for these policies.

3. **It is simple and computationally efficient.** AdvAA proposed the use of GAN to solve for the maximization of the minimum loss function, which requires additional training costs for the adversarial network. In contrast, the proposed LatentAugment can solve the max-min problem using the conditional probability ($\tilde{h}_z^{(t)}$) of the highest loss without an adversarial network. LatentAugment can be solved using a simple stochastic gradient descent

Table 1: Comparing the training cost and test accuracy between proposed LatentAugment (LA) with RandAugment (RA (Cubuk et al., 2019)), Adversarial AutoAugment (AdvAA (Zhang et al., 2019)), and Uncertainty-Based Sampling (UBS (Wu et al., 2020)) using the Wide-ResNet 28-10 model on CIFAR-10. Training cost of required GPU hours is reported relative to RA.

|  | RA | AdvAA | UBS | LA ($K=2$) | LA ($K=4$) | LA ($K=6$) |
|---|---|---|---|---|---|---|
| Training cost | 1 | 8 | 1.5 | 1.9 | 3.3 | 4.7 |
| Test accuracy | 95.8% | 98.1% | 97.9% | 98.0% | 98.2% | 98.3% |

algorithm. Table 1 compares the training cost of required GPU hours between proposed LatentAugment and other methods.

4. **It is a general model which includes other augmentation methods.** The proposed LatentAugment is a general augmentation method that includes other methods such as UBS and AdvAA. Following theorem shows that the UBS with a single data point and AdvAA could be considered to be special cases of LatentAugment:

**Theorem 2.1.** *(Special Case of LatentAugment). Assume that the unconditional probabilities for all augmentation policies are the same ($\pi_z = 1/S, \forall z$). If the inverse scale parameter $\sigma \to 0$, the gradient of expected loss function of the LatentAugment can be equal to the one of UBS. If $\sigma \to \infty$, the gradient of expected loss function of the adversarial network with LatentAugment is equal to the one of AdvAA.*

The proof can be found in Appendix A.2. Note that the the gradient of expected loss function of LatentAugment with $\sigma \to \infty$ can be equivalent to the one of AdvAA. However, it means that LatentAugment can not maximize minimum loss without an additional network, thus the advantages in efficiency of LatentAugment will be lost.

## 3 EXPERIMENTS

### 3.1 EXPERIMENT SETTING

This section describes the experiments investigating the performance of the proposed LatentAugment using the CIFAR-10 and CIFAR-100 (Krizhevsky & Hinton, 2009), SVHN (Netzer et al., 2011), and ImageNet (Russakovsky et al., 2015) datasets. In these experiments, the network models are Resnet-50 (He et al., 2016), Wide-ResNet 40-2 and Wide-ResNet 28-10 (Zagoruyko & Komodaki, 2016), Shake-Shake 26 2×32d, 26 2×96d, and 26 2×112d (Gastaldi, 2017), and PyramidNet with ShakeDrop with a depth of 272 and an alpha of 200 (Han et al., 2017; Yamada et al., 2018). Table 4 in the Appendix A.4 provides the hyperparameters. All the hyperparameters of the network models are the same as those used in AA (Cubuk et al., 2018), Fast AA (Lim et al., 2019), and PBA (Ho et al., 2019). A cosine learning decay with one annealing cycle was applied to all models.

As proposed by Wu et al. (2020), we use 16 transformations: AutoContrast, Brightness, Color, Contrast, Cutout, Equalize, Invert, Mixup, Posterize, Rotate, Sharpness, ShearX, ShearY, Solarize, TranslateX, and TranslateY. Shorten & Khoshgoftaar (2019) described these transformations. As in AA, augmentation policies are generated by the combination of the two transformations. The size of the policy set was $S = 16 \times 16 = 256$. Following AA, we set the magnitude of all the transform operations in a moderate range. All values of the magnitude range are same as AA.

The unconditional probability ($\pi_z$) was initialized as $1/S$. The range of the magnitude of each transformation was discretized into 10, which were randomly drawn from the uniform distribution. The unconditional probability ($\pi_z$) was calculated using the moving average. The length of the moving average was fixed at 10 iterations in this experiment. It was difficult to estimate the expected loss function using the full set $\mathbb{S}$ of augmentation policies because of the computational burden of the large size $S = 256$. Alternatively, subset $\mathbb{K}$ could be used which is randomly drawn from the full set $\mathbb{S}$. In this experiment, the subset size of the augmentation policies was set to six ($K = 6$). The inverse scale parameter $\sigma$ was set to one. The effects of the unconditional probability, subset size, and inverse scale are discussed in later sections of the paper.

### 3.2 CIFAR-10 RESULTS

The CIFAR-10 dataset has a total of 60,000 images, including 50,000 for the training set and 10,000 for the test set. Each image with a size of 32 ×32 belongs to one of the 10 classes. The baseline is trained with standard data augmentation using horizontal flips with 50% probability, zero-padding, and random crops. The proposed LatentAugment first applies the baseline preprocessing, then applies LatentAugment using six policies randomly drawn from 256 policies, and finally applies the Cutout (DeVries & Taylor, 2017) or the Cutmix (Yun et al., 2019).

Table 2 shows the results of the test accuracy for different network models using the CIFAR-10 dataset. For all models, the proposed LatentAugment method achieved a better performance com-

Table 2: Test accuracy (%) on CIFAR-10, CIFAR-100, SVHN, and ImageNet. All experiments in this study replicate the results of Baseline and AutoAugment (AA) (Cubuk et al., 2018), Adversarial AutoAugment (AdvAA) (Zhang et al., 2019), Uncertainty-Based Sampling (UBS) (Wu et al., 2020), and MetaAugment (MA) (Zhou et al., 2020). On the proposed LatentAugment (LA), averages of five runs are reported. Network models are Wide-ResNet 40-2 and Wide-ResNet 28-10 (Zagoruyko & Komodaki, 2016), Shake-Shake 26 2×32d, 26 2×96d, and 26 2×112d (Gastaldi, 2017), PyramidNet with ShakeDrop (Han et al., 2017; Yamada et al., 2018) and Resnet-50 (He et al., 2016). See text for more details.

| Dataset | Model | Baseline | AA | AdvAA | UBS | MA | LA |
|---|---|---|---|---|---|---|---|
| CIFAR-10 | WRN40-2 | 94.70 | 96.30 | - | - | 96.79 | **97.27±0.09** |
| | WRN28-10 | 96.13 | 97.32 | 98.10 | 97.89 | 97.76 | **98.25±0.08** |
| | S-S (26 2x32d) | 96.45 | 97.53 | 97.64 | - | - | **97.68±0.03** |
| | S-S (26 2x96d) | 97.14 | 98.01 | 98.15 | 98.27 | 98.29 | **98.42±0.02** |
| | S-S (26 2x112d) | 97.18 | 98.11 | 98.22 | - | 98.28 | **98.44±0.02** |
| | PyramidNet | 97.33 | 98.52 | 98.64 | 98.66 | 98.57 | **98.72±0.02** |
| CIFAR-100 | WRN40-2 | 74.00 | 79.30 | - | - | 80.60 | **80.90±0.15** |
| | WRN28-10 | 81.20 | 82.91 | 84.51 | 84.54 | 83.79 | **84.98±0.12** |
| | S-S (26 2x96d) | 82.95 | 85.72 | 85.90 | - | **85.97** | 85.88±0.10 |
| SVHN | WRN28-10 | 98.50 | 98.93 | - | - | - | **98.96±0.01** |
| ImageNet | ResNet-50 (Top 1) | 75.30 | 77.63 | 79.40 | - | 79.74 | **80.02±0.10** |
| | ResNet-50 (Top 5) | 92.20 | 93.82 | 94.47 | - | 94.64 | **94.88±0.05** |

pared to existing augmentation methods. For example, LatentAugment achieved an improvement of 0.15% and 0.36% compared to AdvAA and UBS on Wide-ResNet 28-10 model, respectively. The test accuracy of the proposed LatentAugment using PyramidNet+ShakeDrop was 98.72 %, which was 0.08% and 0.06% better than that of AdvAA and UBS, respectively. To compare with the AA, we tested the proposed model using the same transformations as AA, which uses the policy set with SamplePairing (Inoue, 2018), instead of Mixup (Zhang et al., 2017), and finally applied the Cutout, instead of Cutmix. The test accuracies for the Wide-ResNet 40-2 model and Wide-ResNet 28-10 of the LatentAugment using the same transformation as AA were 96.91±0.05% and 98.01±0.05%, respectively (Table 5). Thus, the proposed method outperforms AA, even when neither Mixup nor Cutmix were used. Adversarial AutoAugment (AdvAA) also applies the same transformations as AA, although the subset size of AdvAA is 8. To compare with AdvAA, we tested the model using the same subset size. The test accuracy of Wide-ResNet 28-10 of the LatentAugment using the same transformations as AA with subset size $K = 8$ is 98.16±0.07%. Thus, the proposed method is marginally better than AdvAA even when the same transformations and subset sizes are used. See table 5 and 6 in Appendix.

## 3.3 CIFAR-100 RESULTS

The CIFAR-100 dataset also has a total of 60,000 images, including 50,000 for the training set and 10,000 for the test set. The number of categories is 100. The procedure of the baseline and LatentAugment is the same as that of CIFAR-10.

As for CIFAR-10, the proposed LatentAugment indicated better accuracy than existing augmentation methods except Shake-Shake (26 2×96d), in which the test accuracy of LatentAugment was slightly lower than that of AdvAA and MetaAugment (MA).

## 3.4 SVHN RESULTS

The SVHN dataset has 73,257 digit images for the core training set, 531,131 for the additional training set, and 26,032 for the test set. In this experiment, both core and additional training sets were used. The number of categories is 10. The baseline was trained using the normalizing data. The proposed method first applies LatentAugment using six policies, randomly drawn from 256 policies, then normalizes the data, and finally applies the cutout with a region size of 20 ×20 pixels,

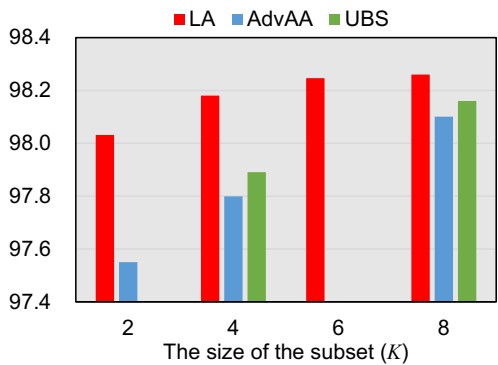

Figure 2: The test accuracies with the different size of the subset ($K$). It shows the test accuracies of LatentAugment (LA), Uncertainty-Based Sampling (UBS), Adversarial AutoAugment (AdvAA) with the different size of the subset ($K$) using the Wide-ResNet 28-10 model on CIFAR-10. This figure replicates the results of UBS from Wu et al. (2020) and AdvAA from Zhang et al. (2019).

following the method proposed by DeVries & Taylor (2017). LatentAugment using Wide-ResNet 28-10 achieves 0.03% improvement compared to AA.

### 3.5   IMAGENET RESULTS

The ImageNet dataset has more than 1.2 million training images, 50,000 validation images, and 100,000 test images. The number of categories is 1,000. Following the AA, baseline augmentation uses the standard Inception-style pre-processing, including horizontal flips with 50% probability and random distortions of colors. The proposed LatentAugment first applies the baseline preprocessing, then applies LatentAugment using six policies randomly drawn from 256 policies, and finally applies the Cutmix. The proposed method outperformed previous augmentation studies.

### 3.6   CHOICE OF THE SUBSET SIZE

This experiment used a subset size of $K = 6$. To determine the optimal size of the subset, this study used the Wide-ResNet 28-10 to evaluate the performance of the proposed LatentAugment with different $K$, where $K \in \{2, 4, 6, 8\}$. Figure 2 suggests that the test accuracy of the model rapidly increases up to $K = 6$. However, no significant improvement was observed when $K$ was 8. In contrast, the computational cost increases with $K$. Therefore, after comparing the computational cost and performance, all the experiments in this study used $K = 6$ for LatentAugment.

Figure 2 also shows the results of AdvAA and UBS. AdvAA uses instances of $K \in \{2, 4, 8, 16, 32\}$ for each input example, augmented by adversarial policies. The study of UBS reports the experimental results using $K = 4$ with a single data point and $K = 8$ with four data points for training. This figure suggests that the proposed LatentAugment is more efficient than AdvAA and UBS, because LatentAugment with $K = 4$ outperforms AdvAA and UBS with $K = 8$.

### 3.7   THE EFFECTS OF THE INVERSE SCALE

LatentAugment requires determining the inverse scale parameter $\sigma$ which is assumed to have a value of 1 in the previous section. This section considers the effects of the inverse scale using different values. Figure 3 shows the test accuracy of LatentAugment with different inverse scale values using the Wide-ResNet 40-2 model on CIFAR-10. This suggests that the test accuracy is maximum at $\sigma = 1$, although the effect of the inverse scale is weak except for $\sigma \to 0$.

### 3.8   THE EFFECTS OF THE UNCONDITIONAL PROBABILITY

LatentAugment estimates the unconditional probabilities ($\pi_s$) as well as the network parameters ($\theta$). As shown in Theorem 2.1, if the unconditional probabilities are fixed at the same values, the

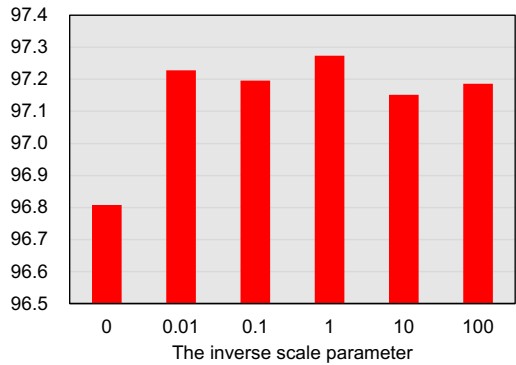

Figure 3: The test accuracies with the different inverse scale parameter ($\sigma$). It shows the test accuracies of LatentAugment with the different inverse scale values using the Wide-ResNet 40-2 model on CIFAR-10.

Table 3: The test accuracies using fixed or unfixed unconditional probability with the different inverse scale parameters. Averages of five runs are reported.

| | Inverse scale parameter ($\sigma$) | | Diff. |
| | $\sigma = 0$ | $\sigma = 1$ | |
| --- | --- | --- | --- |
| Fixed $\pi_z$ | (a) 96.71±0.10 | (b) 97.19±0.08 | 0.48 |
| Unfixed $\pi_z$ | (c) 96.81±0.10 | (d) 97.27±0.09 | 0.47 |
| Diff. | 0.10 | 0.08 | |

derivative of LatentAugment can be reduced to that of adversarial AA or UBS. Table 3 shows the effects of the unconditional probability in LatentAugment using the Wide-ResNet 40-2 model on CIFAR-10. The cell of (a), where the unconditional probabilities ($\pi_z$) are fixed and the inverse scale parameter ($\sigma$) is set to 0, is equivalent to UBS with a single data point of the highest loss. In contrast, the cell of (d), where $\pi_z$ can be estimated and $\sigma = 1$, is the test accuracy of the proposed LatentAugment, which allows variable $\pi_z$ and multiple data points for the expectation of the loss function. This table suggests that an unfixed $\pi_z$ can slightly improve the test accuracy over a fixed $\pi_z$. However, the effect on test accuracy with an fixed $\pi_z$ is weaker than the effect of $\sigma$ set to zero. Thus, for the better performance of the proposed LatentAugment, using multiple data points for the expected value of the loss function weighted by the conditional probability of the highest loss, has a more significant effect on the performance than using the unfixed unconditional probability.

## 4 CONCLUSIONS

This study introduces LatentAugment, which estimates the probability of the latent augmentation customized to each input image and network model. The proposed method is appealing in that it can dynamically optimize the augmentation methods for each input and model parameter in learning iterations. As shown in the theoretical analysis, LatentAugment is a general model that includes AdvAA and UBS as special cases. Furthermore, the proposed method is simple and computationally efficient in comparison with the existing methods, which requires a generative adversarial network. Experimental results show that the proposed LatentAugment has better performance than previous augmentation methods on the CIFAR-10, CIFAR-100, SVHN, and ImageNet datasets. Finally, an open question remains in the robustness of LatentAugment using the EM algorithm, which typically converges to a local optimum. While we checked the stability of the test accuracy with five runs using different random seeds, the issue of convergence of LatentAugment is an important theme for future research. An application to the object detection, image generation, and text recognition using LatentAugment is also an interesting topic. We leave such directions to future work.

## 5 REPRODUCIBILITY STATEMENT

Pytorch code of the experiments in this paper can be downloaded from GitHub (`https://github.com/xxxx/xxxxxx`).

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

# A APPENDIX

## A.1 RANDOMLY DRAWN SUBSET OF THE AUGMENTATION POLICIES

Let $\delta_z$ be the probability that policy $z$ can be drawn. The conditional probability with $\delta_z$ can be written as:

$$h_z^{(t)} = \frac{\pi_z^{(t)} P\left(y \mid o_z\left(x\right), \theta^{(t)}\right)}{\sum\limits_{z \in \mathbb{S}} \delta_z \pi_z^{(t)} P\left(y \mid o_z\left(x\right), \theta^{(t)}\right)}.$$

The expected loss function using the randomly drawn subset given $\delta_z$ is

$$\mathcal{E}\left(\Theta|\Theta^{(t)}, \delta_z\right) = -\mathbb{E}_{(x,y) \sim (X,Y)} \sum_{z \in \mathbb{S}} \delta_z h_z^{(t)} \log\left(\pi_z^{(t)} P\left(y \mid o_z\left(x\right), \theta\right)\right)$$

$$= -\mathbb{E}_{(x,y) \sim (X,Y)} \sum_{z \in \mathbb{S}} \frac{\delta_z \pi_z^{(t)} P\left(y \mid o_z\left(x\right), \theta^{(t)}\right)}{\sum\limits_{z \in \mathbb{S}} \delta_z \pi_z^{(t)} P\left(y \mid o_z\left(x\right), \theta^{(t)}\right)} \log\left(\pi_z^{(t)} P\left(y \mid o_z\left(x\right), \theta\right)\right).$$

Assume that the policies of the subset are drawn using simple random draws: $\delta_z = \delta, \forall z \in \mathbb{S}$. Under this assumption, the expected loss function using a randomly drawn subset is equal to the expected loss function using the full set:

$$\mathcal{E}\left(\Theta|\Theta^{(t)}, \delta\right) = -\mathbb{E}_{(x,y) \sim (X,Y)} \sum_{z \in \mathbb{S}} \frac{\delta \pi_z^{(t)} P\left(y \mid o_z\left(x\right), \theta^{(t)}\right)}{\delta \sum\limits_{z \in \mathbb{S}} \pi_z^{(t)} P\left(y \mid o_z\left(x\right), \theta^{(t)}\right)} \log\left(\pi_z^{(t)} P\left(y \mid o_z\left(x\right), \theta\right)\right) = \mathcal{E}\left(\Theta|\Theta^{(t)}\right).$$

## A.2 PROOF OF THEOREM 2.1 (SPECIAL CASE OF LATENTAUGMENT)

### A.2.1 UNCERTAINTY-BASED SAMPLING (UBS)

The loss function of UBS is

$$\mathcal{L}_{UBS} = \mathbb{E}_{(x,y) \sim (X,Y)} \max_{z \in \mathbb{K}} \left[-\log\left(P\left(y \mid o_z\left(x\right), \theta\right)\right)\right].$$

If the inverse scale $\sigma \to 0$, the conditional probability $\tilde{h}_z$ can be approximated by the indicator function:

$$\lim_{\sigma \to 0} \tilde{h}_z = \lim_{\sigma \to 0} \frac{\exp\left(-\frac{h_z}{\sigma}\right)}{\sum\limits_{r \in \mathbb{K}} \exp\left(-\frac{h_r}{\sigma}\right)} = I\left[h_z \leq h_r, \forall r \in \mathbb{K}\right],$$

where $I\left[\cdot\right]$ is the indicator function such that $I\left[\cdot\right] = 1$ if $h_z \leq h_r, \forall r \in \mathbb{K}$, and $I\left[\cdot\right] = 0$, otherwise.

If $\pi_z = 1/S$ for all $z$ and $\sigma \to 0$ in the LatentAugment, $\tilde{h}_z \to I\left[h_z \leq h_r, \forall r \in \mathbb{K}\right] = I\left[P\left(y \mid o_z\left(x\right), \theta^{(t)}\right) \leq P\left(y \mid o_r\left(x\right), \theta^{(t)}\right), \forall r \in \mathbb{K}\right]$. Therefore, the expected loss function of the LatentAugment is

$$\tilde{\mathcal{E}}\left(\Theta|\Theta^{(t)}\right) \to -\mathbb{E}_{(x,y) \sim (X,Y)} \mathbb{E}_{\mathbb{K} \sim \mathbb{S}}$$

$$\left[\sum_{z \in \mathbb{K}} I\left[P\left(y \mid o_z\left(x\right), \theta^{(t)}\right) \leq P\left(y \mid o_r\left(x\right), \theta^{(t)}\right), \forall r \in \mathbb{K}\right] \log\left(P\left(y \mid o_z\left(x\right), \theta\right)\right) + \log\left(1/S\right)\right],$$

$$\tilde{\mathcal{E}}\left(\Theta|\Theta^{(t)}\right)\Big|_{\theta = \theta^{(t)}} \to \mathbb{E}_{(x,y) \sim (X,Y)} \mathbb{E}_{\mathbb{K} \sim \mathbb{S}} \max_{z \in \mathbb{K}} \left[-\log\left(P\left(y \mid o_z\left(x\right), \theta^{(t)}\right)\right)\right] - \log\left(1/S\right).$$

Note that the first term is the same as the loss function of the uncertainty-based sampling evaluated at $\theta = \theta^{(t)}$, while the second term is constant. Therefore, $\nabla_\theta \tilde{\mathcal{E}}\left(\Theta|\Theta^{(t)}\right)\Big|_{\theta = \theta^{(t)}} \to \nabla_\theta \mathcal{L}_{UBS}|_{\theta = \theta^{(t)}}$, if $\pi_z = 1/S$ for all $z$ and $\sigma \to 0$.

### A.2.2 ADVERSARIAL AUTOAUGMENT (ADVAA)

The loss function of AdvAA is

$$\mathcal{L}_{AdvAA} = -\mathbb{E}_{(x,y)\sim(X,Y)}\mathbb{E}_{\mathbb{K}\sim\mathcal{A}(\mathbb{S},\mu)}\left[\frac{1}{K}\sum_{z\in\mathbb{K}}\log\left(P\left(y\,|\,o_z\left(x\right),\theta\right)\right)\right],$$

where $\mathcal{A}\left(\mathbb{S},\mu\right)$ is the adversarial network for the set of augmentation policies, $\mathbb{S}$ with parameter $\mu$. If $\sigma \to \infty$ in the LatentAugment, $\tilde{h}_z \to 1/K$. Assume $\pi_z = 1/S$ for all $z$ in LatentAugment. Then, the loss function of the adversarial network with LatentAugment is equal to the loss function of AdvAA plus constant:

$$\mathbb{E}_{\mathbb{K}\sim\,\mathcal{A}(\mathbb{S},\mu)}\left[\tilde{\mathcal{E}}\left(\Theta|\Theta^{(t)}\right)\right] \to -\mathbb{E}_{(x,y)\sim(X,Y)}\mathbb{E}_{\mathbb{K}\sim\mathcal{A}(\mathbb{S},\mu)}\left[\frac{1}{K}\sum_{z\in\mathbb{K}}\log\left(P\left(y\,|\,o_z\left(x\right),\theta\right)\right)\right]-\log\left(1/S\right).$$

Therefore, $\nabla_\theta\mathbb{E}_{\mathbb{K}\sim\mathcal{A}(\mathbb{S},\mu)}\left[\tilde{\mathcal{E}}\left(\Theta|\Theta^{(t)}\right)\right] \to \nabla_\theta\mathcal{L}_{AdvAA}$, if $\pi_z = 1/S$ for all $z$ and $\sigma \to \infty$.

### A.3 COMPUTER RESOURCES

We train the models with the LatnetAugment using computers with 4 NVIDIA RTX 2080Ti GPUs and 64 GB memory.

### A.4 HYPERPARAMETERS

Table 4: Hyperparameters for the experiment. LR represents the learning rate, whereas WD represents the weight decay.

| Dataset | Model | LR | WD | Batch Size | Epoch |
|---------|-------|-----|-----|------------|-------|
| CIFAR-10 | Wide-ResNet-40-2 | 0.1 | 0.0002 | 128 | 200 |
| CIFAR-10 | Wide-ResNet-28-10 | 0.1 | 0.0005 | 128 | 200 |
| CIFAR-10 | Shake-Shake (26 2x32d) | 0.01 | 0.001 | 128 | 1800 |
| CIFAR-10 | Shake-Shake (26 2x96d) | 0.01 | 0.001 | 128 | 1800 |
| CIFAR-10 | Shake-Shake (26 2x112d) | 0.01 | 0.001 | 128 | 1800 |
| CIFAR-10 | PyramidNet+ShakeDrop | 0.05 | 5E-05 | 64 | 1800 |
| CIFAR-100 | Wide-ResNet-40-2 | 0.1 | 0.0002 | 128 | 200 |
| CIFAR-100 | Wide-ResNet-28-10 | 0.1 | 0.0005 | 128 | 200 |
| CIFAR-100 | Shake-Shake (26 2x96d) | 0.01 | 0.0025 | 128 | 1800 |
| SVHN | Wide-ResNet-28-10 | 0.005 | 0.001 | 128 | 160 |
| ImageNet | Resnet-50 | 0.1 | 0.0001 | 256 | 270 |

### A.5 TEST ACCURACIES USING THE SAME TRANSFORMATIONS AS AA AND ADVAA.

This section provides comparison between proposed LatentAugment (LA) and AutoAugment (AA) or Adversarial AutoAugment (AdvAA) using same subset size and transformations. To compare with the AA, we tested the proposed model using the same transformations as AA, which uses the policy set with SamplePairing (Inoue, 2018), instead of Mixup (Zhang et al., 2017), and finally applied the Cutout, instead of Cutmix. Table 5 shows the test accuracies for the Wide-ResNet 40-2 model and Wide-ResNet 28-10 of the LA using the same transformation as AA. This table also provides the result of UBS using same transformations as AA, reported by Wu et al. (2020).

AdvAA applies the same transformations as AA, although the subset size of AdvAA is 8. To compare with AdvAA, we tested the model using the same subset size. Table 6 shows the test accuracy of Wide-ResNet 28-10 of the LA using the same transformations as AA with subset size $K = 8$.

Table 5: Test accuracies (%) on CIFAR-10 using the same transformations as AutoAugment (AA). On the proposed LatentAugment (LA), averages of five runs are reported.

| Model | AA | UBS | LA ($K = 6$) |
|---|---|---|---|
| WRN40-2 | 96.30 | - | 96.91±0.05 |
| WRN28-10 | 97.32 | 97.75 | 98.01±0.05 |

Table 6: Test accuracies (%) on CIFAR-10 using the same subset size and transformations as Adversarial AutoAugment (AdvAA). On the proposed LatentAugment (LA), an average of five runs is reported.

| Model | AdvAA | LA ($K = 8$) |
|---|---|---|
| WRN28-10 | 98.10 | 98.16±0.07 |

### A.6 TRANSFERABILITY ACROSS DATASETS AND ARCHITECTURES

This section evaluates transferability with LatentAugment across different datasets and model architectures. We first take a snapshot of the unconditional probability $\pi_z$ of ResNet-50 on ImageNet using LA, and then apply the fixed $\pi_z$ to train the models of Wide-ResNet 40-2 on CIFAR-10 or CIFAR-100 using LA. Table 7 provides the experimental result of transferability. It suggests that LA with policy transfer has still good performance.

Table 7: The test accuracies of the transfer the unconditional probability of augmentation policies.

| Dataset | Baseline | LA (direct) | LA (policy transfer) |
|---|---|---|---|
| CIFAR-10 | 94.7 | 97.27±0.09 | 97.21±0.09 |
| CIFAR-100 | 74.0 | 80.90±0.15 | 80.77±0.14 |

### A.7 ADDITIONAL EXPERIMENTS USING OTHER DATASETS

This section provides the results of additional experiments using datasets of MNIST (LeCun et al., 1998), Fashion MNIST (Xiao et al., 2017), and Oxford flowers102 (Nilsback & Zisserman, 2008).

#### A.7.1 MNIST

The MNIST is a large database of handwritten digits. It has a total of 70,000 images, including 60,000 for the training set and 10,000 for the test set. Each example is a 28x28 grayscale image, associated with a label from 10 classes. The baseline is trained with standard data augmentation using zero-padding, and random crops. The proposed LatentAugment first applies the baseline preprocessing, then applies LatentAugment using six policies randomly drawn from 256 policies, and finally applies the Cutout. We use hyperparameters of WRN40-2 on CIFAR-10 in Table 4.

#### A.7.2 FASHION MNIST

The Fashion MNIST is a dataset of Zalando's article images—consisting of a training set of 60,000 examples and a test set of 10,000 examples. Each example is a 28x28 grayscale image, associated with a label from 10 classes. The baseline is trained with standard data augmentation using horizontal flips with 50% probability, zero-padding, and random crops. The proposed LatentAugment first applies the baseline preprocessing, then applies LatentAugment using six policies randomly drawn from 256 policies, and finally applies the Cutmix. We use hyperparameters of WRN40-2 on CIFAR-10 in Table 4

### A.7.3 OXFORD FLOWERS102

Oxford 102 Flower is an image classification dataset consisting of 102 flower categories. The flowers chosen to be flower commonly occurring in the United Kingdom. Each class consists of between 40 and 258 images. Baseline augmentation uses the standard Inception-style pre-processing, including horizontal flips with 50% probability and random distortions of colors. The proposed LatentAugment first applies the baseline preprocessing, then applies LatentAugment using six policies randomly drawn from 256 policies, and finally applies the Cutmix. We use hyperparameters of ResNet-50 on ImageNet in Table 4

Table 8: The test accuracies of the additional experiments using MNIST (LeCun et al., 1998), Fashion MNIST (Xiao et al., 2017), and Oxford flowers102 (Nilsback & Zisserman, 2008).

| Dataset | Model | Baseline | LA |
|---|---|---|---|
| MNIST | WRN40-2 | 99.66 | 99.77±0.02 |
| Fashion MNIST | WRN40-2 | 94.54 | 96.04±0.06 |
| Flowers102 | ResNet-50 | 95.70 | 98.19±0.09 |

### A.8 CONVERGENCE OF LOSS FUNCTIONS

Convergence of the EM algorithm is usually defined as a sufficiently small change in the loss function (Aitkin & Aitkin, 1996). To confirm the convergence, Figure 4 shows the loss functions using the network models of Wide-ResNet 40-2, Wide-ResNet 28-10, Pyramid, and Shake-Shake on CIFAR-10 and CIFAR-100. This figure indicates the convergence of an estimation using the EM algorithm.

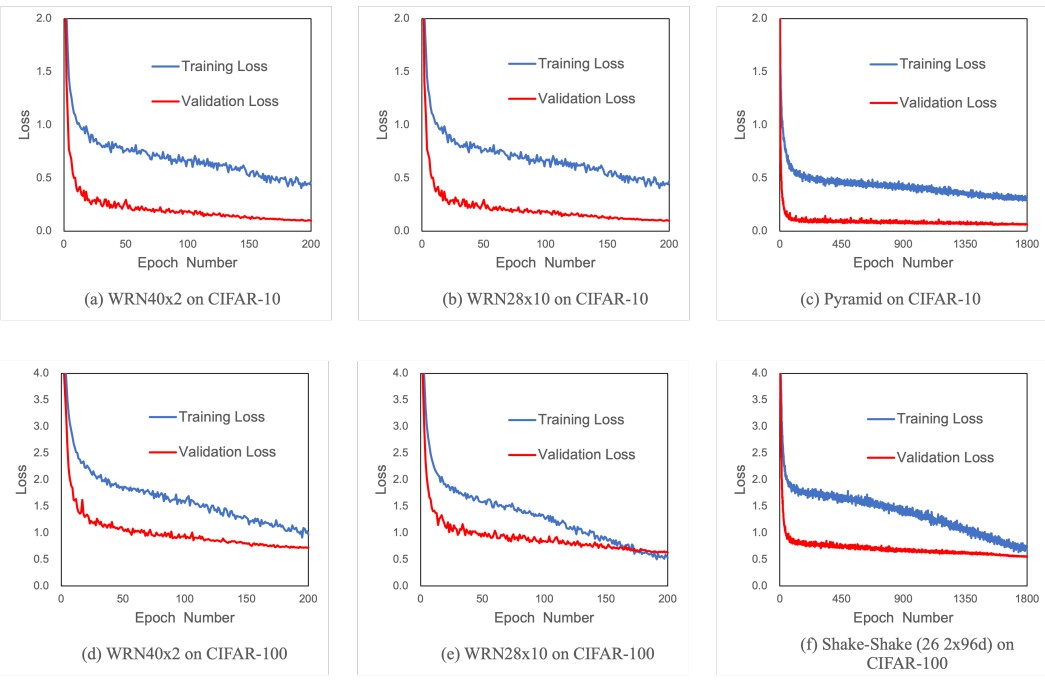

Figure 4: The loss functions of the different network models on CIFAR-10 and CIFAR-100.

