# OpenReview forum: "LatentAugment: Dynamically Optimized Latent Probabilities of Data Augmentation"
_ICLR.cc/2023/Conference — Submitted to ICLR 2023_

### Official Review · Reviewer_AEKr · 2022-10-23

**Confidence:** 2
**Correctness:** 2
**Technical Novelty And Significance:** 2
**Empirical Novelty And Significance:** 2
**Recommendation:** 5

**Clarity, Quality, Novelty And Reproducibility:**

The paper is written very clearly. However, the originality of the work should be more elaborated.

**Strength And Weaknesses:**

Strengths:

1. The paper is well-organized, with each section clearly stating what it does. The contribution of the paper has been clearly outlined.

2. Experiments look sound and justify the main claim of the paper.

Weaknesses:

1. The level of novelty is relatively low. On a high-level, LatentAugment appears to be a straightforward application of the EM algorithm over the logarithmic losses defined with respect to conditional probability. The intuition behind the choice of loss function is also a bit vague. It would be helpful to explain why softmax is used and how the conditional probability relates to the goal of the LatentAugment algorithm.

2. Except for the classification task, no other task has been used to test the performance of LatentAugment. It would be helpful to incorporate other relevant tasks where data augmentation is helpful (e.g. inpainting, image generation) to see how LatentAugment is useful in general.

3. CIFAR, SVHN, and ImageNet are public datesets very commonly used across different image tasks. Has there been any tests on other less commonly used datasets where LatentAugment might be helpful?

**Summary Of The Paper:**

This paper proposes a new framework for data augmentation that generalizes other existing data augmentation methods by incorporating Bayesian inference methods. Empirical performances on common benchmark datasets suggest that the LatentAugment methods outperforms other Adversarial benchmarks not only as measured by test accuracy in classification tasks, but also by training cost.

**Summary Of The Review:**

My personal evaluation would be a 5. I'd be happy to adjust my score, if the authors respond to the concerns accordingly.

---

> ### Author Response · Authors · 2022-11-17
> **Response to reviewer AEKr**
>
> Thank you for your very valuable comments. We have addressed some of your comments below:
>
> > The level of novelty is relatively low. On a high-level, LatentAugment appears to be a straightforward application of the EM algorithm over the logarithmic losses defined with respect to conditional probability. The intuition behind the choice of loss function is also a bit vague. It would be helpful to explain why softmax is used and how the conditional probability relates to the goal of the LatentAugment algorithm.
>
> The novelty of proposed LatentAugment (LA) is not only an application of the EM algorithm to data augmentation, but also **LA does not require an additional network to search the optimal augmentation policies.** As shown in Adversarial AutoAugment (AdvAA) [1], maximizing the minimum loss can improve the performance of test accuracy avoiding the overlearning problem. AdvAA uses an additional adversarial network. In contrast, LA can solve the max-min problem without an additional network.
>
> **Our originality is using the $\tilde{h}_z$ instead of $h_z$ in the EM algorithm**. As shown in equation (3), the conditional probability $\tilde{h}_z$ is the probability such that augmentation policy $z$ has the highest loss. Using $\tilde{h}_z$, LA can maximize the minimum loss without an additional network.
>
> Following table compares between the test accuracy estimated model using $h_z$ and $\tilde{h}_z$. We use the network model of Wide-ResNet 40-2 on CIFAR-10 dataset. It suggests that the standard EM algorithm (LA with $h_z$) has less performance due to overlearning.
>
> | Baseline | LA with $h_z$ | LA wit $\tilde{h}_z $ |
> |:----:|:----:|:----:|
> |94.70|96.78$\pm$0.06|**97.27$\pm$0.09**|
>
>
> The transformation function from $h_z$ to $\tilde{h}_z$ is **softmin** function, not **softmax**. We have corrected it in the revised manuscript. While softmax function is used to calculate the probability of argmax $Pr[h_z \ge h_k \forall k]$, softmin function calculate the probability of argmin $Pr[h_z \le h_k \forall k]$. From the definition of $\mathcal{L}_z$, the probability of minimum $h_z$ is equal to the one of maximum $\mathcal{L}_z $. **Thus, the softmin function is related to the goal of the LA, maximization of minimum loss.**
> To clarify, we have added the description of the relation between softmin and the goal of LA, after equation (3).
>
> > Except for the classification task, no other task has been used to test the performance of LatentAugment. It would be helpful to incorporate other relevant tasks where data augmentation is helpful (e.g. inpainting, image generation) to see how LatentAugment is useful in general.
>
> Thank you for valuable suggestion. Although we focused on the image classification task, we consider that LA might have good performance in other tasks. Wu et al. [2] extends UBS to text augmentation and reports that UBS outperforms the previous studies. **As shown in the theorem 2.1, UBS is the special case of LA. Therefore, LA might have also good performance in text augmentation.** Following your suggestion, we added the description of applicability of other tasks in the Conclusion section.
>
> > CIFAR, SVHN, and ImageNet are public datesets very commonly used across different image tasks. Has there been any tests on other less commonly used datasets where LatentAugment might be helpful?
>
> Following your suggestion, **we tried additional experiments using other datasets.** The results are as follows:
>
> | Dataset | Baseline | LA |
> |:----:|:----:|:----:|
> | MNIST | 99.66 | **99.77$\pm$0.02** |
> | Fashion MNIST | 94.54 | **96.04$\pm$0.06** |
> | Flowers102 | 95.70 |**98.19$\pm$0.09** |
>
> We added subsection A.7 that provides additional experiments in the revised paper. **The results show that LA has also good performance for other datasets.** Thank you for your informative suggestion.
>
> Thanks once again Reviewer AEKr for your valuable insight. I hope that we have satisfactorily addressed your concerns.
>
> [1] Zhang, X. et al. Adversarial AutoAugment. In ICLR, 2019.
>
> [2] Wu, S. et al. On the generalization effects of linear transformations in data augmentation. In ICML, 2020.

---

### Official Review · Reviewer_M9Bp · 2022-10-23

**Confidence:** 3
**Correctness:** 3
**Technical Novelty And Significance:** 3
**Empirical Novelty And Significance:** 3
**Recommendation:** 8

**Clarity, Quality, Novelty And Reproducibility:**

The paper is well-written and the idea is novel to me.  Code is provided in the supplementary (but I do not have chance to test the code).

**Strength And Weaknesses:**

[Strength]

The paper is well-written and easy to follows.

Novelty and with theoretical analysis provided.

The model looks simple and computational efficiency, but would be good if with evidence supported.

The results are encouraging with ablation study provided.

[Weakness]

The claim on computational efficiency needs to be supported with theoretical or empirical results. For example, it would improve a paper if the authors can provide the computational comparison to other methods.

**Summary Of The Paper:**

The paper addresses the learn optimal policy for data augmentation on image classification task.  Learning the optimal augmentation policy, which is the latent variable, interesting research problem. The paper proposes a simple via EM-based method (call LatentAugment) to estimate the probability of optimal policy via conditional probability of the policy customized for given input and network model. It also provides the theorical analysis to show some exisitng methods (AdvAA and UBS) as the special cases. Experiments are on CIFAR-10, CIFAR-100, SVHN, and ImageNet datasets using different network architectures showing the encouraging results that it outperforms most of existing methods.

**Summary Of The Review:**

Overall, I think the method is novel to me and most of claims are well-supported except one in the weakness. The paper is in the form can be published.

---

> ### Author Response · Authors · 2022-11-17
> **Response to reviewer M9Bp**
>
> Thank you for your very positive comments. We have addressed some of your comments below:
>
> > The claim on computational efficiency needs to be supported with theoretical or empirical results. For example, it would improve a paper if the authors can provide the computational comparison to other methods.
>
> Thank you for great suggestion. Theoretical results suggest that proposed LatentAugment (LA) more efficient than Adversarial AutoAugment (AdvAA)[1], that requires additional adversarial network. In contrast, LA can solve the max-min problem without an additional network.
> Experimental results support the theoretical analysis. Table 1 shows compares the training cost of required GPU hours between LA and other methods. This table suggests that **LA is more efficient in the training cost than AdvAA.** Although Uncertainty-Based Sampling (UBS)[2] is also efficient, UBS shows slightly less performance in test accuracy.
>
> For the memory efficiency, the required GPU memory of LA, using the Wide-ResNet 28-10 model on CIFAR-10, is as follows. Unfortunately, we cannot compare the memory efficiency between LA and other models, because the authors of AdvAA and UBS did not report the required GPU memory.
>
> | Baseline | LA ($K=2$ ) | LA ($K=4$ ) | LA ($K=6$ ) |
> |:----:|:----:|:----:|:----:|
> |5.3GB|9.0GB|13.9GB|18.7GB|
>
> > The paper is well-written and the idea is novel to me. Code is provided in the supplementary (but I do not have chance to test the code).
>
> Thank you for your positive comment. Although we have submitted the Pytorch code in the supplementary file in the OpenReview, we added **the Reproducibility Statement section** for the code in the revised paper. Following your suggestion, we will upload the code to GitHub after the review process.
>
> Thanks once again Reviewer M9Bp for your valuable insight. I hope that we have satisfactorily addressed your concerns.
>
> [1] Zhang, X. et al. Adversarial AutoAugment. In ICLR, 2019.
>
> [2] Wu, S. et al. On the generalization effects of linear transformations in data augmentation. In ICML, 2020.

---

### Official Review · Reviewer_AC1v · 2022-10-25

**Confidence:** 2
**Correctness:** 4
**Technical Novelty And Significance:** 2
**Empirical Novelty And Significance:** 2
**Recommendation:** 5

**Clarity, Quality, Novelty And Reproducibility:**

I believe the proposed method has good reproducibility and some novelty, while the quality of the paper needs to be improved. Specifically, I believe more experiments are needed, and the presentation needs to be polished.

**Strength And Weaknesses:**

Strength:

The proposed LatentAugment is straightforward and reasonable;

Compared to some previous methods, the proposed method is efficient in terms of computation cost;

Weakness:

The performance gain is relatively small. According to Table 1, UBS is also an efficient method, while the proposed method is only slightly better than UBS with more computation cost. Given the results, I'm curious about result of K=1 for Table 1, from which we may better compare the training cost and performance of proposed method and UBS.

Given that the results are slightly better than previous method, more experiments are needed. For example, instead of vanilla fully-supervised learning, how will the proposed method benefit some settings where data augmentations might be really important, including few-shot learning, transfer learning, meta learning, etc.

Some presentation needs to be improved. For instance, Figure 2 looks incomplete because some bars seems to be missing without clear explanation. I would suggest the authors either compared all the related methods under all the same settings, or explain clearly about the figure/results.




**Summary Of The Paper:**

The paper proposed a simple yet effective data augmentation method, which is called LatentAugment.

**Summary Of The Review:**

Although the paper propose a simple and effective method, I believe the paper needs improvement before getting accepted.

---

> ### Author Response · Authors · 2022-11-17
> **Response to reviewer AC1v**
>
> Thank you for valuable comments and implications. We have modified the paper following your comments.
>
> > The performance gain is relatively small. According to Table 1, UBS is also an efficient method, while the proposed method is only slightly better than UBS with more computation cost.
>
> Although uncertainty-based sampling (UBS)[1] is efficient method, Adversarial AutoAugment (AdvAA) [2] has better performance in accuracy than UBS. In contrast, the proposed LatentAugment (LA) outperforms the AdvAA and is more efficient than AdvAA. Furthermore, **LA is more general model than UBS.** As shown in the Theorem 2.1, UBS is a special case of LA.
>
> > Given the results, I'm curious about result of K=1 for Table 1, from which we may better compare the training cost and performance of proposed method and UBS.
>
> Although LA with $K=1$ is more efficient, it has less performance. If $K=1$, equation (4) suggests $\tilde{h}_z = 1$ and $\pi_z = 1/S$. Thus, the LA cannot maximize minimum loss, and it will be reduced to the standard model with randomly drawn augmentation policies.
>
> > Given that the results are slightly better than previous method, more experiments are needed.
>
> A.6 in Appendix provides the transfer learning with the LA. Table 7 suggests that LA with policy transfer has still good performance. Following your suggestion, **we tried additional experiments using other datasets.** The results are as follows:
>
>
> | Dataset | Baseline | LA |
> |:----:|:----:|:----:|
> | MNIST | 99.66 | **99.77$\pm$0.02** |
> | Fashion MNIST | 94.54 | **96.04$\pm$0.06** |
> | Flowers102 | 95.70 |**98.19$\pm$0.09** |
>
> We added subsection A.7 that provides additional experiments in the revised paper. **The results show that LA has also good performance for other datasets.** Thank you for your informative suggestion.
>
> > Some presentation needs to be improved. For instance, Figure 2 looks incomplete because some bars seems to be missing without clear explanation.
>
> Some bars in Figure 2 are missing, because the authors of AdvAA and UBS did not report the values. As shown in the text of page 8, AdvAA uses instances of $K\in \lbrace2, 4, 8, 16, 32 \rbrace$, while UBS reports the results of $K=4$ with a single data point and $K=8$ with four data points.
>
> > I would suggest the authors either compared all the related methods under all the same settings, or explain clearly about the figure/results.
>
> To compare with the AutoAugment (AA)[3], we tested the proposed model using the same transformations as AA (Table 5). This table also shows the result of UBS using same transformations as AA. To compare with AdvAA, we tested the model using the same subset size and transformations as AdvAA (Table 6). Following tables suggest that **the proposed LA outperforms AA, UBS, and AdvAA, even when same transformations were used.** Although we have provided these results in subsection 3.2, we added these tables in the section A5 in Appendix.
>
> Table 5. Test accuracies (%) on CIFAR-10 using the same transformations as AA.
> | Model| AA | UBS | LA ($K=6$ ) |
> |:----:|:----:|:----:|:----:|
> |WRN40-2|96.30|-|**96.91$\pm$ 0.05**|
> |WRN28-10|97.32|97.75|**98.01$\pm$0.05**|
>
> Table 6. Test accuracies (%) on CIFAR-10 using the same subset size and transformations as AdvAA.
> | Model| AdvAA | LA ($K=8$ ) |
> |:----:|:----:|:----:|
> |WRN28-10|98.10|**98.16 $\pm$ 0.07**|
>
>
>
> Thanks once again Reviewer AC1v for your valuable insight. I hope that we have satisfactorily addressed your concerns.
>
>
> [1] Wu, S. et al. On the generalization effects of linear transformations in data augmentation. In ICML, 2020.
>
> [2] Zhang, X. et al. Adversarial AutoAugment. In ICLR, 2019.
>
> [3] Cubuk, E. D. et al. AutoAugment: Learning augmentation policies from data. In CVPR, 2018.

---

### Official Review · Reviewer_LaAK · 2022-10-25

**Confidence:** 4
**Correctness:** 3
**Technical Novelty And Significance:** 3
**Empirical Novelty And Significance:** 3
**Recommendation:** 6

**Clarity, Quality, Novelty And Reproducibility:**

\- Clarity: the presentation of the paper should be improved in several aspects.
In the introduction I consider that the authors should not limit the presentation of previous methods to AutoAugment, even if the other approaches are mentioned in related work. I do not see many similarities between the proposed work and the Bayesian augmentations of Tran et al. (2017). Fig. 1 has a disconnect with the proposed formulation. For instance in Fig. 1 the E-step is applied on the multiplication of the conditional probabilities of the policies and the losses, while in Equ. 4 the conditional probabilities are multiplied by $P(y|o_z(x),\theta)$. In method, in the third line the authors mention the use of random augmentations, but then $\pi_z$ is used. In 2.1 first equation is presented as Bayes' rule, but in my understanding is just probability normalization. Algorithm 1 contains too much text and some definitions of variables are inaccurate. Equ. 3 seems a bit strange as it normalizes $h_z$ which is based on already normalized probabilities, thus twice exponentiation which might not be ideal for gradient propagation.

\-Quality: the method seems to outperform previous approaches, however the differences are relatively small. It would be important to provide the code in order to make sure that the improvements are due to the algorithm and not different and better hyperparameters.

\- Novelty: as mentioned previously, the paper seems very similar to advAA, thus the novelty is limited. Authors should explicitly present the improvements with respect advAA in related work as well with experiments.

\- Reproducibility: authors provide the hyperparameters for the trained model. However, it would be important also to provide the code and make sure that the obtained results are due to only the algorithm and not different hyper-parameters than previous work.

**Strength And Weaknesses:**

\+ The proposed approach seems to generate more meaningful augmentations than previous approaches as shown in the experimental results

\- Different parts of the paper are disconnected and difficult to follow. See clarity for more details.

\- The differences between this work and adversarial autoaugment (advAA) seem minimal. As shown by theorem 2.1, with uniform unconditional probabilities and $\sigma \rightarrow \infty$ the proposed approach is advAA. As shown in table 3, the contribution of considering variable unconditional probabilities is quite low and in the order of the std. Not sure about the difference between $\sigma = 1$ and a high value. It would be interesting to see table 3.

\- Differences in classification accuracy are relatively small. How can we verify that those differences are not due to a larger batch sizes or other implementation details

**Summary Of The Paper:**

This paper presents a new algorithm for data augmentation based on the same transformations used for autoaugment and follows-up papers. The algorithm considers policies that are composed of two sequential transformations selected from 16. Thus in total there are 256 different policies that can be selected. The probabilities of selecting those policies are latent variables that are estimated during training with a expectation maximization approach. In addition the algorithm can estimate also the probabilities of a policy conditioned to the given sample by normalizing the corresponding losses. Results show that the proposed algorithm outperforms previous approaches in most of the cases.

**Summary Of The Review:**

I consider this paper a valuable contribution for ICLR. However there are several points that the authors should improve for acceptance.

\- the writing the paper seems rushed, with some parts that are disconnected and other that should be improved and clarified (see clarity).
\- the differences with advAA seems minor. Authors should present and evaluate the differences with advAA.

---

> ### Author Response · Authors · 2022-11-17
> **Response to reviewer LaAK (Part 1)**
>
> Thank you for your interest in our paper and your feedback. We will address your very important concerns below:
>
> > The differences between this work and adversarial autoaugment (advAA) seem minimal. As shown by theorem 2.1, with uniform unconditional probabilities and $\sigma \to \infty$ the proposed approach is advAA.
>
> Adversarial AutoAugment (AdvAA) [1] requires additional adversarial network to maximizing minimum loss. In contrast, **the proposed LatentAugment (LA) does not require any additional network for searching the augmentation policy.** Therefore, LA is more efficient than AdvAA. Theorem 2.1 suggests that the gradient of expected loss function of LA can be equivalent to the one of AdvAA, if $\pi_z = 1/S$ and $\sigma \to \infty$. However, for this case, LA cannot maximize minimum loss without an additional network, thus the advantages in efficiency of LA will be lost. To clarify, we added the description of the difference between AdvAA and LA after the theorem in the revised paper (page 6).
>
> > Differences in classification accuracy are relatively small. How can we verify that those differences are not due to a larger batch sizes or other implementation details
>
> As show in Table 4, all hyperparameters of our experiments, including batch sizes, are same as those of the previous studies. To compare with the AutoAugment (AA) [2], we tested the proposed model using the same transformations as AA (Table 5). This table also shows the result of Uncertainty-Bases Sampling (UBS) [3] using same transformations as AA. To compare with AdvAA, we tested the model using the same subset size and transformations as AdvAA (Table 6). Following tables suggest that **the proposed LA outperforms AA, UBS, and AdvAA, even when same transformations were used.** Although we have provided these results in subsection 3.2, we added these tables in the section A5 in Appendix.
>
> Table 5. Test accuracies (%) on CIFAR-10 using the same transformations as AA.
> | Model| AA | UBS | LA ($K=6$ ) |
> |:----:|:----:|:----:|:----:|
> |WRN40-2|96.30|-|**96.91$\pm$ 0.05**|
> |WRN28-10|97.32|97.75|**98.01$\pm$0.05**|
>
> Table 6. Test accuracies (%) on CIFAR-10 using the same subset size and transformations as AdvAA.
> | Model| AdvAA | LA ($K=8$ ) |
> |:----:|:----:|:----:|
> |WRN28-10|98.10|**98.16 $\pm$ 0.07**|
>
>
> > Fig. 1 has a disconnect with the proposed formulation. For instance in Fig. 1 the E-step is applied on the multiplication of the conditional probabilities of the policies and the losses, while in Equ. 4 the conditional probabilities are multiplied by $ P( y|o_z( x),\theta)$.
>
> As show in subsection 2.2, the loss using the augmentation policy $z$ is defined as $\mathcal{L}_z = - \log(\pi_z \cdot P(y|o_z( x),\theta))$.
>
> The multiplication of the conditional probability of the policy and the losses is $h_z \mathcal{L}_z = - h_z \log(\pi_z) - h_z \log(P(y|o_z( x),\theta))$, as equation 4. Therefore, Fig. 1 is consistent with the model.
>
>
> > In method, in the third line the authors mention the use of random augmentations, but then $\pi_z$ is used.
>
> You are correct. As you pointed out, the proposed LA does not use random data augmentation but randomly drawn subset from augmentation policies. Following your suggestion, we have deleted this line. Thank you for your valuable comment.
>
>
> > In 2.1 first equation is presented as Bayes' rule, but in my understanding is just probability normalization.
>
> The conditional probability (**posterior**) is $h_z(z|y) = \pi_z P(y|z)  / P(y)$, where $\pi_z$ is unconditional probability (**prior**), $ P(y|z)$ is the likelihood, and $P(y)=\sum \pi_z P(y|z)$. To clarify, I added the reference [4] for this equation.
>
> [1] Zhang, X. et al. Adversarial AutoAugment. In ICLR, 2019.
>
> [2] Wu, S. et al. On the generalization effects of linear transformations in data augmentation. In ICML, 2020.
>
> [3] Cubuk, E. D. et al. AutoAugment: Learning augmentation policies from data. In CVPR, 2018.
>
> [4] McLachlan, G. J. and T. Krishnan. The EM algorithm and extensions, John Wiley & Sons, 2007.

---

> > ### Author Response · Authors · 2022-11-19
> > **Response to reviewer LaAK (Part 2)**
> >
> > > Equ. 3 seems a bit strange as it normalizes hz which is based on already normalized probabilities, thus twice exponentiation which might not be ideal for gradient propagation.
> >
> > Thank you for important comments. For the M-step, **the expected loss function with fixed $\tilde{h}_z$ is minimized**. Therefore, the gradient is not affected by the twice exponentiation. To clarify, we added the description before equation (5).
> >
> >
> > > Algorithm 1 contains too much text and some definitions of variables are inaccurate.
> >
> > Following your comments, we have revised the Algorithm 1. Thank you for valuable suggestion.
> >
> >
> > > Quality: the method seems to outperform previous approaches, however the differences are relatively small. It would be important to provide the code in order to make sure that the improvements are due to the algorithm and not different and better hyperparameters.
> >
> > Although we have submitted the Pytorch code in the supplementary file in the OpenReview, following your suggestion, we added the **Reproducibility Statement section** for the code in the revised paper. We will upload the code to GitHub after the review process.
> >
> >
> > > the differences with advAA seems minor. Authors should present and evaluate the differences with advAA
> >
> > Following your comments, we have added the description of the difference between AdvAA and LA  after the theorem in the revised paper (page 6). Thank you for your informative suggestion.
> >
> > Thanks once again Reviewer LaAK for your valuable insight. I hope that we have satisfactorily addressed your concerns.

---

### Author Response · Authors · 2022-11-23
**Response to all reviewers**

We appreciate the reviewers for constructive feedback on our paper. We present here our response to some questions that are asked by multiple reviewers.

**Reproducibility Statement section**

We have submitted the Pytorch code in the supplementary file in the OpenReview. Following your suggestion, we added the Reproducibility Statement section for the code in the revised paper.

**Additional experiments using other datasets**

Following your suggestion, we tried additional experiments using other datasets. We added subsection A.7 that provides additional experiments in the revised paper. The results show that the proposed LatentAugment (LA) has also good performance for other datasets.

**Comparison using the same transformations as other methods**

We tested the proposed model using the same transformations as AutoAugment (AA) (Table 5). To compare with Adversarial AutoAugment (AdvAA), we tested the model using the same subset size and transformations as AdvAA (Table 6). The results suggest that the proposed LA outperforms other methods. We have added these tables in the section A5.

**Clarification of originality of the proposed method**

The novelty of proposed LA is not only an application of the EM algorithm to data augmentation. LA does not require an additional network to search the optimal augmentation policies. AdvAA requires additional adversarial network to maximizing minimum loss. In contrast, LA does not require any additional network, therefore, LA is more efficient in the training cost than AdvAA. Furthermore, LA is general model including other methods, as shown in the theorem 2.1. We have added the description of originality of the proposed method and differences from other methods in the revised manuscript.

We thank all the reviewers for their valuable comments and suggestions.

---

### Decision · Program_Chairs · 2023-01-20

**Decision:**

Reject

**Justification For Why Not Higher Score:**

Positive but modest empirical improvements; issues with clarity that lead to some confusion around the formulation.

**Justification For Why Not Lower Score:**

N/A

**Metareview: Summary, Strengths And Weaknesses:**

This paper introduces a data augmentation formulation which treats the optimal augmentation policy as a latent variable to be inferred. The empirical results for this approach improve results on a number of classification baselines. It also has a strong advantage that it is computationally cheap, and simple and straightforward to implement.

That said, reviews were borderline in part because the improvements were rather modest; suggestions from reviewers include exploring the use of data augmentation in other tasks beyond classification. There were also concerns from some reviewers regarding novelty relative to existing approaches such as AdvAA, as well as clarity of the paper and how the method connects to related work. This extends to understanding the overall formulation; the paper claims to be finding an optimal policy through EM, but it is not expanded on what notion of optimality is meant here, and in practice the objective function which is used is quite different from EM due to the use of the "softmin" in equation (3) rather than the posterior in equation (2). This is expanded upon below, in the summary of the AC-reviewer meeting.

**Summary Of Ac-Reviewer Meeting:**

This was a borderline paper and led to a discussion over email between the reviewers. Ultimately, this led to the decision to reject the paper, as none of the reviewers were willing to champion the paper and argue strongly for its acceptance.

Importantly we also felt that some of the questions around the formulation that were raised by Reviewer LaAK were not adequately addressed. The Bayesian formulation of data augmentation taken here has little to do with the Bayesian augmentation of Tran et al (2017) cited, and in fact is a different setup. However although the paper focuses on the EM algorithm for inferring the optimal policy, the actual approach described in the algorithm and in figure 1 is a bit different. Maybe a more direct motivation and understanding of the $\tilde h_z^{(t)}$ quantities would help in a revision of the paper.